# Transcriptome Changes Reveal the Molecular Mechanisms of Humic Acid-Induced Salt Stress Tolerance in Arabidopsis

**DOI:** 10.3390/molecules26040782

**Published:** 2021-02-03

**Authors:** Joon-Yung Cha, Sang-Ho Kang, Myung Geun Ji, Gyeong-Im Shin, Song Yi Jeong, Gyeongik Ahn, Min Gab Kim, Jong-Rok Jeon, Woe-Yeon Kim

**Affiliations:** 1Division of Applied Life Science (BK21four), Plant Molecular Biology and Biotechnology Research Center, Research Institute of Life Sciences, Gyeongsang National University, Jinju 52828, Korea; mgj2930@gnu.ac.kr (M.G.J.); shin123@gnu.ac.kr (G.-I.S.); songyi123@gnu.ac.kr (S.Y.J.); 2Genomics Division, National Institute of Agricultural Sciences, Rural Development Administration, Jeonju 54874, Korea; hosang93@korea.kr; 3Department of Agricultural Chemistry and Food Science & Technology, Institute of Agriculture and Life Science, Gyeongsang National University, Jinju 52828, Korea; ahngi@gnu.ac.kr (G.A.); jrjeon@gnu.ac.kr (J.-R.J.); 4College of Pharmacy and Research Institute of Pharmaceutical Science, Gyeongsang National University, Jinju 52828, Korea; mgk1284@gnu.ac.kr

**Keywords:** arabidopsis, humic acid, salt stress, transcriptome analysis

## Abstract

Humic acid (HA) is a principal component of humic substances, which make up the complex organic matter that broadly exists in soil environments. HA promotes plant development as well as stress tolerance, however the precise molecular mechanism for these is little known. Here we conducted transcriptome analysis to elucidate the molecular mechanisms by which HA enhances salt stress tolerance. Gene Ontology Enrichment Analysis pointed to the involvement of diverse abiotic stress-related genes encoding HEAT-SHOCK PROTEINs and redox proteins, which were up-regulated by HA regardless of salt stress. Genes related to biotic stress and secondary metabolic process were mainly down-regulated by HA. In addition, HA up-regulated genes encoding transcription factors (TFs) involved in plant development as well as abiotic stress tolerance, and down-regulated TF genes involved in secondary metabolic processes. Our transcriptome information provided here provides molecular evidences and improves our understanding of how HA confers tolerance to salinity stress in plants.

## 1. Introduction

Humic acid is composed of a complex supramolecular association known as humic substances (HSs), which is produced by humification of organic matter such as peat, compost, and plants in soil environments [1]. Although HSs are amorphous and depend on their source of extraction from soil, partial polymeric structures can be identified containing diverse aromatic and aliphatic structures [2,3]. These physicochemical properties have not only prompted their agricultural use as soil amendments, but also as plant growth stimulators [3,4,5]. There is evidence that wheat seedlings directly take up and accumulate HA in tissues [6], suggesting that HA stimulates diverse transcriptional changes promoting the physiological and developmental processes of plants. 

Salinity stress is one of the major abiotic stresses reducing crop yield, afflicting 20% of the total arable land worldwide [7]. Soil salinity affects the morphological, physiological, and biochemical processes of plants across all developmental stages, including seed germination, root and vegetative growth, and production of cereals and fruits [7]. High salt levels accompany additive cellular stresses such as ionic, osmotic, and oxidative stress in plants [8,9]. Ionic stress, mainly caused by Na^+^ ions, is prevented by maintaining cellular ionic homeostasis, including levels of Na^+^, K^+^, and Ca^2+^ ions [8]. Various transporters localized in the plasma membrane and the tonoplast are directly involved in redistribution and compartmentation of ions. To protect against increases in osmotic potential, plants accumulate metabolites functioning as osmolytes, such as soluble sugars, glycine betaine, and proline [10]. Salt-induced oxidative stress in plants triggers membrane damage, and reactive oxygen species (ROS) are scavenged by antioxidant enzymes [9,10]. In promoting these responses, salinity stress turns on multiple genes, and drives multiple stress signaling networks and physiological mechanisms enhancing plant tolerance. 

We found that HA enhances tolerance to excessive salt levels via post-transcriptional control of the HIGH-AFFINITY K^+^ TRANSPORTER 1 in Arabidopsis [11]. Most recently, we also reported that HA drives global transcriptome changes, especially genes involved in response to heat including genes encoding HEAT-SHOCK PROTEINs (HSPs) [12], which are essential for acquiring thermotolerance in Arabidopsis [13]. Although numerous biological effects of HA and HSs have been reported in plants [5], the precise transcriptome analysis based on HA-induced stress tolerance has not been identified. 

Here we elucidate the transcriptome changes due to HA application under salt stress conditions to understand how HA confers salt stress tolerance in transcriptional levels. Under salt stress conditions, HA up-regulates genes involved in response to stimuli, such as genes encoding HSPs and GLUTAREDOXINs (GRXs/ROXYs) and down-regulates genes involved in response to biotic stress and metabolic processes. We also identify the transcripts commonly regulated by HA both in the absence and presence of salt stress. These findings suggest that the wide-ranging roles of HA promote agricultural use for a plant growth stimulator and a protective agent helping endure against unfavorable environments.

## 2. Results and Discussion

### 2.1. HA Application Confers Salt Stress Yolerance in Arabidopsis and Italian Ryegrass

We recently reported that HA promotes seed germination, lateral root development, and salt stress tolerance in Arabidopsis [11,14]. Here we reevaluated the HA-induced salt stress tolerance at the germination stage of Arabidopsis, unlike previous salt stress response monitored at the seedling stage. Arabidopsis (Columbia ecotype background) wild-type seeds were germinated in Murashige and Skoog (MS) media or MS containing 860 mg L^−1^ HA supplemented with or without 100 mM NaCl. Seed germination and consequent early seedling development were suppressed by salinity stress, as previously reported [15], but its inhibition was partly rescued by HA application (Figure 1a). Relative fresh weight and chlorophyll content (in the presence vs. absence of 100 mM NaCl) were significantly higher in HA applied plants (Figure 1b,c). Thus, these data and our previous finding suggest that HA helps escape salt-induced inhibition of seed germination as well as that of seedling development in Arabidopsis. 

To extend the agricultural application of HA, we examined HA-induced salt tolerance in Italian ryegrass, a high-quality forage crop widely cultivated in temperate regions. We previously found that HA promotes seedling growth and regrowth after cutting of Italian ryegrass by foliar application [16]. Ten-d old Italian ryegrass seedlings were exposed to salt stress by submerging in 250 mM NaCl solution, and HA (860 mg L^−1^) or water (as a control) applied directly by foliar spraying at 0, 3, and 6 days after salt treatments. Under 250 mM NaCl treatments for 10 d, HA-sprayed seedlings were extremely tolerant compared to water-sprayed seedlings (Figure 1d). Fresh weight and chlorophyll content were significantly higher in HA-treated Italian ryegrass seedlings compared to water-treated seedlings (Figure 1e,f). Interestingly, these results indicate that foliar application of HA helps plants overcome the soil-born salt stress through an unknown pathway and, in addition, bioactivity of HA including salt stress tolerance is not limited to Arabidopsis, a model plant. HA thus stimulates defense systems of various plants under salt stress. 

### 2.2. Illumina Sequencing, Mapping Sequence Reads, and Total Differentially Expressed Genes (DEGs)

HSs positively affect both soil properties and plants, and HA is suggested to modulate biochemical and molecular processes involving nutrient uptake, lateral root development, stomatal response, and salt tolerance at both the transcriptional and post-transcriptional level [17,18,19,20]. In a previous study, we reported the transcriptome changes by HA application in Arabidopsis, and found that 21% of 416 differentially expressed genes (DEGs) up-regulated by HA enriched in the “response to stimulus” Gene Ontology (GO) category to “ biological processes” (GO:0008150) [12]. To elucidate the molecular evidence of HA-induced salt stress tolerance, we performed transcriptome analysis using Arabidopsis seedlings exposed to salt (100 mM NaCl) or salt + HA (860 mg L^−1^) treatment for 9 h with previous HA-treated samples [12]. 

We sequenced cDNA libraries constructed for each treatment using the Illumina HiSeq 2500 platform. A total of 632,280,782 sequence reads were produced from all four treatments (MS as a control, HA, salt, and salt + HA) including treatments in a previous report (MS or HA treatment [12]) with three replications. On average, 94% of the quality-filtered reads were generated for all samples uniquely mapped to the reference genome. The other reads were either 4% unmapped or did not show primary hits (2%). A summary of mapping statistics obtained for each treatment is described in Appendix A. 

Transcriptome changes in Arabidopsis seedlings subjected to four treatments were analyzed, and the total transcriptome from each treatment was compared to identify the DEGs among pairs of treatments. From comparisons between MS and salt (DEG #1), MS and HA (DEG #2), salt and salt + HA (DEG #3), and HA and salt + HA (DEG #4), a total of 3785, 3257, 5271, and 2582 DEGs were isolated, respectively (Appendix A). In a previous report, we identified DEG #2, mainly regulated by HA, and found that HA triggers transcriptional activation of genes encoding HEAT-SHOCK PROTEINs (HSPs) and consequently confers heat stress tolerance in Arabidopsis [12]. Thus, in this study, we focused on the transcriptional changes in DEG #3 to elucidate the molecular mechanisms of HA conferring salt stress tolerance in Arabidopsis. 

### 2.3. Up-Regulated Genes by HA under Salt Stress (DEG #3)

Of 5271 DEGs (log_2_ fold change ≥ 0.3 and cutoff *q*-value < 0.05) in DEG #3, 2483 genes were significantly up-regulated by HA, while 2788 genes were significantly down-regulated by HA (Appendix A). As described for DEG #2 [12], we re-analyzed DEGs showing a log_2_ fold change ≥1 and ≤−1 for up- and down-regulated genes, respectively. Under salt stress conditions, a total of 515 genes were up-regulated by HA application, while 620 genes were down-regulated by HA. GO term enrichment analysis [21] was performed for the 515 up-regulated genes in DEG #3, and the 50 most up-regulated genes in DEG #3 are shown in Appendix A. Among the 515 genes, 22% of DEGs were significantly enriched (False Discovery Rate, FDR = 1.5 × 10^−6^) in the “response to stimulus” (GO:0050896) GO category (Figure 2a and Appendix A). Among these, 34 genes were associated with the “response to abiotic stimulus” (GO:00009628, FDR = 1.8 × 10^−5^), 41 genes associated with the “response to stress” (GO:0006950, FDR = 0.00038), and 33 genes associated with “response to chemical stimulus” (GO:0042221, FDR = 0.012) (Figure 2b).

#### 2.3.1. Up-regulated Genes Involved in Response to Light Stimulus

Genes associated with the “response to light stimulus” were *PHYTOCHROME KINASE SUBSTRATE 4* (*PKS4*, AT5G04190), *FAR-RED-ELONGATED HYPOCOTYL1-LIKE* (*FHL*, AT5G02200), *LONG HYPOCOTYL IN FAR-RED* (*HFR1*, AT1G02340), *INDOLE-3-ACETIC ACID INDUCIBLE 29* (*IAA29*, AT4G32280), *LIGHT-HARVESTING CHLOROPHYLL A/B-BINDING 2.3* (*LHCB2.3*, AT3G07500), *ARABIDOPSIS THALIANA HOMEOBOX PROTEIN 16* (*ATHB16*, AT4G40060), and *GLUTAMATE SYNTHASE 1* (*GLU1*, AT5G04140). Red light receptor phytochromes (phy) regulate plant growth including seed germination and hypocotyl elongation [22]. *FAR-RED ELONGATED HYPOCOTYL* (*FHY*) and FHL transmit the phyA signal to their downstream transcription factors (TFs) *HFR1* and *LONG AFTER FAR-RED LIGHT 1* (*LAF1*), inhibiting hypocotyl elongation [23]. *FHY* and *FAR1* up-regulated by ABA and abiotic stresses, acting as positive regulators of ABA and integrators of light and ABA signaling [24]. One of the four PKS proteins in Arabidopsis, PKS4, negatively regulates phyA/B-mediated hypocotyl growth inhibition and is involved in phototropism [25]. GLU1 is a FERREDOXIN-DEPENDENT GLUTAMINE OXOGLUTARATE AMINOTRANSFERASE 1 (Fd-GOGAT1) catalyzing the synthesis of glutamate from glutamine and α-ketoglutarate, and is involved in growth and development [26]. LHCB proteins are the most abundant chloroplast proteins in plants and mainly function in collecting and transferring light energy to photosynthetic reaction centers, positively affecting plant development as well as stress tolerance [27]. Auxin-inducible *IAA* genes promote plant development, and IAA29 is activated by PHYTOCHROME-INTERACTING FACTOR 4 (PIF4) binding to control morphological acclimation to warm temperature [28]. Salt stress causes numerous harmful effects, such as toxic ROS induction and photoinhibition, resulting in the retardation of plant growth [29]. In the presence of salt stress, HA may help to maintain light absorption, light-harvesting capacity, and light-induced plant growth, and subsequently enhance plant survivability. 

#### 2.3.2. Up-Regulated Genes Involved in Response to Heat

Genes enriched in the “response to heat” (GO:0009408, FDR = 0.0008) GO term were also associated with GO terms “response to temperature stimulus” (GO:0009266) and “response to stress” (GO:0006950) (Figure 2b). *HSP101* (AT1G74310)*,* cytosolic *HSP81.1* (AT5G52640), *CLASS I SMALL HSP17.6* (AT1G53540), *CLASS II SMALL HSP17.6* (AT5G12020), *SMALL HSP 17.6A* (AT5G12030), *MITOCHONDRION-LOCALIZED SMALL HSP23.6* (*HSP23.6-MITO*, AT4G25200), *HS TRANSCRIPTION FACTOR A3* (*HSFA3*, AT5G03720), *TEMPERATURE-INDUCED LIPOCALIN* (*TIL*, AT5G58070), and *RESPIRATORY BURST OXIDASE HOMOLOG B* (*ATRBOHB*, AT1G09090) up-regulated by HA in DEG #3. Small HSPs with a monomer size of 10–40 kDa form large oligomers and function as molecular chaperones to prevent thermal denaturation of substrates [30]. The transcripts of small HSPs are rapidly induced by stress, and plants overexpressing small HSPs exhibit not only heat tolerance but also tolerance to diverse abiotic stresses via increased chaperoning capacity resulting in protein homeostasis [31,32]. The HSFA3 transcription factor activates many heat-inducible genes, and its overexpression enhances thermotolerance [33]. TIL is involved in heat and salt stress tolerance, probably by preventing membrane lipid peroxidation caused by heat stress and inhibiting chloroplast destruction caused by ion toxicity, respectively [34,35]. *RBOHB* encodes an NADPH oxidase generating H_2_O_2_ and is specifically expressed in roots; the increased superoxide confers salt stress tolerance in grafted cucumber by promoting Na^+^ exclusion from roots and early stomatal closure [8,36]. Thus, HA may protect salt-induced intracellular protein damage via the transcriptional activation of HSPs and RBOHB. 

#### 2.3.3. Up-Regulated Genes Involved in Response to Cell Redox Homeostasis

In addition, up-regulated genes involved in cell redox homeostasis (GO:0045454, FDR = 0.0178) were significantly enriched in DEG #3 (Figure 2b). These genes encode class III (CC-type) GRXs/ROXYs (AT3G62930, ROXY16; AT4G15680, ROXY13; AT1G15690, ROXY12; AT4G15700, ROXY11; AT5G18600, ROXY10), acting as glutathione-dependent disulfide oxidoreductases involved in oxidative stress responses [37]. ROXY genes are differentially regulated by nitrate; 6 ROXYs (ROXY6, 8, 9, 19–21) are up-regulated and 10 ROXYs (ROXY7, 10–18) are down-regulated under nitrate deprivation, while 7 ROXYs (ROXY4, 11–13, 15–17) are up-regulated by the addition of nitrate [38,39]. A gain-of-function study using ROXY15 suggests a positive and negative involvement in chlorophyll content and root hair elongation, respectively [39]. Thus, HA helps plants escape salt-induced toxicity through induction of ROXY genes to maintain cellular redox homeostasis, which subsequently drive physiological outputs such as proper functioning of chlorophyll and normal root development under salt stress conditions. 

### 2.4. Down-Regulated Genes by HA under Salt Stress (DEG #3)

GO term enrichment analysis [21] of 620 genes down-regulated by HA in DEG #3 revealed that 65% of these DEGs (409/620) were associated with the “biological process” category (GO:0008150) and significantly enriched (FDR < 0.05) in the “regulation of biological process” (GO:0050789), “multi-organism process”(GO:0051704), and “response to stimulus” (GO:0050896) GO categories (Appendix A). The 50 most down-regulated genes in DEG #3 are shown in Appendix A. The GO categories of down-regulated genes in DEG #3 were biotic stress-related (“response to biotic stimulus”, GO:0009607; “defense response”, GO:0006952; “response to wounding”, GO:0009611) and primary and secondary metabolic process-related (“pigment metabolic process”, GO:0042440; “secondary metabolic process”, GO:0019748; “phenylpropanoid metabolic process”, GO:0009698) GO terms (Figure 3). 

#### 2.4.1. Down-Regulated Genes Involved in Toxin Catabolic Process

Down-regulated genes involved in the “toxin catabolic process” (GO:0009407, FDR < 0.001) were mainly plant-specific phi- (GLUTATHIONE S-TRASNSFERASE F, GSTF) and tau- (GSTU) type GST genes (GSTs; ATGSTU3, AT2G29470; ATGSTU6, AT2G29440; ATGSTU11, AT1G69930; ATGSTU12, AT1G69920; ATGSTF6, AT1G02930; ATGSTF12, AT5G17220). GST catalyzes S-conjugation between the thiol group of reduced glutathione and toxic substrates to reduce cellular toxicity [40]. The expression patterns of GSTs belonging to multi-gene families are differentially regulated by external conditions such as a/biotic stresses and internal factors such as ROS and phytohormones, and their down-regulation by HA under salinity stress might be related to the down-regulation of genes associated with biotic stress-related GO terms [41,42]. 

#### 2.4.2. Down-Regulated Genes Involved in the Metabolic Process

Genes down-regulated by HA in the “metabolic process” category were significantly enriched in the “pigment metabolic process” (GO:0042440, FDR = 0.0011) and “secondary metabolic process” (GO:0019748, FDR = 6.7 × 10^−12^) GO categories (Figure 3b). Genes in the “pigment metabolic process” category were mainly associated with anthocyanin biosynthesis GO terms (“proanthocyanidin biosynthetic process”, GO:0010023; “regulation of anthocyanin biosynthetic process”, GO:0031540; “anthocyanin biosynthetic process”, GO:0009718), such as ANTHOCYANIDIN SYNTHASE (ANS, AT4G22880), PRODUCTION OF ANTHOCYANIN PIGMENT 1 (PAP1, AT1G56650), and TRANSPARENT TESTA 5 (TT5, AT3G55120). Anthocyanins are water-soluble plant pigments of the flavonoid sub-class of phenylpropanoids and are induced by diverse environmental stress responses [43]. In addition, genes enriched in the “secondary metabolic process” GO term mainly belonged to the “phenylpropanoid metabolic process” (GO:0009698, FDR = 2.2 × 10^−12^) and were sub-classified into the “flavonoid metabolic process” (GO:0009812, FDR = 8.7 × 10^−10^) and “phenylpropanoid biosynthetic process” (GO:0009699, FDR = 3.5 × 10^−10^) GO categories. Most anthocyanin-related genes were also associated with both these sub-categories of the “secondary metabolic process” GO term. Flavonoids and phenylpropanoids are secondary metabolites produced in plants, especially for protecting against biotic stress, although they are also involved in abiotic stress responses [44]. 

### 2.5. Validation of DEGs by Quantitative RT-PCR (qRT-PCR)

Transcriptome data were validated by qRT-PCR. Transcripts of three ROXYs including ROXY10, 12, and 13 were enhanced by HA in the presence of salt stress (Figure 4a–c). Two HSP genes such as HSP101 and HSP81.1 increased by salinity stress compared with MS, and furthermore those expressions were stimulated by HA applications in the presence of salt treatment (Figure 4d,e). Interestingly, HSP genes up-regulated by HA both in the absence and presence of salt stress [12], suggesting that HSP genes are major molecular targets of HA facilitating proteostasis. In addition, anthocyanine-involved gene TRANSPARANT TESTA 5 (TT5, AT3G55120) rapidly decreased by HA in the presence of salt stress (Figure 4f). These qRT-PCR data are consistent with transcriptome analysis. 

### 2.6. Transcripts Regulated in Common between DEG #2 (MS vs. HA) and DEG #3 (Salt vs. Salt + HA)

Identification of transcripts commonly expressed between DEG #2 and DEG #3 will help understand the unknown function of HA at the transcriptional level in either the absence or presence of salt stress. Up- and down-regulated DEGs showing log_2_ fold change ≥1 and ≤−1, respectively, were compared, and 269 genes were commonly up-regulated while 188 genes were down-regulated by HA in both DEG #2 and DEG #3 (Figure 5 and Appendix A). We first analyzed the 269 up-regulated genes using GO analysis, and 20% of genes associated with the “biological process” (GO:0008150) category were significantly enriched in the “response to stimulus” (GO:0050896, FDR < 0.05) GO category (Appendix A). Various stress-related genes including those associated with ozone (AT1G01170), osmotic (ALCOHOL DEHYDROGENASE 1 (ADH1), AT1G77120), heat (HSP17.6A, HSP23.6-MITO, and HSP81.1), and universal stress (UNIVERSAL STRESS PROTEIN (USP) FAMILY PROTEINs, AT3G03270 and AT3G62550) responses were up-regulated in both DEG sets. 

In addition, genes involved in the response to oxidative stress (GO:0006979, FDR < 0.05) were significantly enriched: SENESCENCE 1 (SEN1, AT4G35770), HSP17.6A, OXIDATIVE STRESS 3 (OXS3, AT5G56550), ARABIDOPSIS ORTHOLOG OF SUGAR BEET HS1 PRO-1 2 (HSPRO2, AT2G40000), and three unknown genes (AT3G10020, AT5G59080, and AT1G73120). Senescence-associated gene SEN1 was increased by the ROS inducer methyl viologen [45]. OXS3 confers tolerance to heavy metals and oxidative stress, possibly as a chromatic remodeling factor [46]. Heat stress-induced ROS trigger the induction of HSP genes, including HSP17.6A [47]. 

Approximately 30% of all genes encode proteins with currently unknown functions, and genetic transformation has been challenged by identifying the functional roles of these genes. Plants overexpressing AT1G01170 are tolerant to salt stress, while those overexpressing AT5G59080 and AT1G73120 show enhanced tolerance to T-butyl hydroperoxide and paraquat-induced oxidative stress [48,49]. USP genes belong to a large gene family that responds to diverse environmental stresses; USP proteins act as redox-dependent and RNA chaperones, subsequently enhancing heat, cold, and oxidative stress tolerance [50,51]. ADH1 expression is induced substantially by diverse stresses, and plants overexpressing ADH1 show enhanced stress tolerance to salt, drought, cold, and pathogen infection [52]. The analysis of commonly up-regulated genes showed that HA triggers the expression of a variety of genes involved in stress resistance both in the absence and presence of salt stress, conferring a protective effect on plants prior to exposure of stress. 

Among 188 genes down-regulated by HA in both DEG #2 and DEG #3, genes associated with the “metabolic process” (GO:0008152) GO term were significantly enriched in the “pigment metabolic process” (GO:0042440, FDR = 1 × 10^−5^), “secondary metabolic process” (GO:0019748, FDR = 4 × 10^−13^), and “biosynthetic process” (GO:0009058, FDR = 0.0067) GO sub-categories (Figure 5 and Appendix A). The phenylpropanoid-related biosynthetic genes for flavonoids and anthocyanins, PRODUCTION OF ANTHOCYANIN PIGMENT 1 (PAP1, AT1G56650), RIBONUCLEASE 1 (RNS1, AT2G02990), UDP-GLUCORONOSYL/UDP-GLUCOSYL TRANSFERASE (UGT89C1, AT1G06000), TT5, and FLAVONOL SYNTHASE 1 (FLS1, AT5G08640), were down-regulated in both the absence and presence of salt stress. The flavonoid biosynthetic genes determine the testa color, and encoded proteins are involved in various steps for the synthesis of flavonoids and anthocyanidins [53,54]. In transcriptome coexpression analysis in Arabidopsis, UGT89C1 is highly correlated with known flavonoid biosynthetic genes [55]. In addition, genes associated with wounding stress and defense response were down-regulated by HA in both DEG sets: RECEPTOR LIKE PROTEIN 6 (AtRLP6, AT1G45616), AtRLP23 (AT2G32680), AtRLP33 (AT3G05560), THIONIN/PATHOGENESIS-RELATED-13 (PR-13, AT1G66100), LIPOXYGENASE 3 (LOX3, AT1G17420), and DIRIGENT (DIR)-LIKE PROTEIN (AT4G11190). RLPs sensing extracellular signals are transmembrane receptors with extracellular leucine-rich repeat domains, and they play an important role in disease resistance by recruiting RECEPTOR-LIKE KINASE proteins to activate downstream signals [56]. The 13-Lipoxygenase (13-LOX) protein encoded by LOX3 induces oxylipin biosynthesis, which is triggered by the recognition of the avirulent protein Avr-Rpm1 [57]. DIRs and DIR-like protein dictate the stereoselectivity of phenoxy radical coupling during lignin biosynthesis and are transcriptionally induced and spatially targeted during the response to pathogen infection [58,59]. Upon the down-regulation of these positive regulators to biotic stress by HA, HA may negatively modulate the biotic stress response of plants. 

### 2.7. Transcription Factors Regulated in Common between DEG #2 (MS vs. HA) and DEG #3 (Salt vs. Salt + HA)

TFs regulate gene expression in diverse plant development as well as stress responses, and the Arabidopsis genome contains around 2000 TF genes [60]. In total, 27 TF genes were up-regulated by HA in DEG #2 while 44 TF genes up-regulated in DEG #3 (Figure 6a–c and Appendix A). From this, we analyzed the TF genes regulated in common between DEG #2 and DEG #3, and 14 TF genes encoding a bZIP TF (bZIP63/BZO2H3, AT5G28770), CCCH zinc finger TFs (AtTZF/ZFP1 (AT2G25900) and TZF5 (AT5G44260)), B-box type zinc finger TF (BBX17, AT1G49130), ETHYLENE RESPONSE FACTOR (ERF) type TF (HYPOXIA RESPONSIVE ERF 2 (HRE2/ERF71), AT2G47520), ERF TF (AT3G60490), homeodomain-like TF (BROTHER OF LUX ARRHYTHMO (BOA), AT5G59570), GATA type TF (GATA TF 4 (GATA4), AT3G60530), GRAS family TF (AT3G46600), LATERAL ORGAN BOUNDARIES (LOB) DOMAIN-CONTAINING PROTEIN 40 (LBD40, AT1G67100), MADS-box TF (AGAMOUS-LIKE 44 (AGL44), AT2G14210), Myb-like TF (REVEILLE 1 (RVE1), AT5G17300), NUCLEAR FACTOR Y subunit C4 (NF-YC4, AT5G63470), and sequence-specific DNA binding TF (HYPOXIA RESPONSE ATTENUATOR 1 (HRA1), AT3G10040) were commonly up-regulated. AtTZF/ZFP1 is up-regulated by salt stress, and plants overexpressing AtTZF/ZFP1 show enhanced salt stress tolerance by maintaining ionic balance through negative regulation of oxidative and osmotic stresses [61]. TZF5 interacts with the stress-responsive protein RD21A, regulating stress responses [62]. The AP2/ERF family TF gene HRE2/ERF71 is highly responsive to diverse abiotic stresses, such as anoxia, NaCl, mannitol, ABA, and methyl viologen, and transgenic plants overexpressing HRE2/ERF71 are tolerant to stress conditions [63]. The GARP family TF BOA is regulated by the circadian clock, and overexpression of BOA triggers physiological and developmental changes such as increased vegetative growth [64]. GATA4 is involved in the positive regulation of primary and lateral root development [65]. LOB domain-containing genes have 40 homologs with functional redundancy in Arabidopsis producing morphological changes in lateral organ development; LBD40 is mainly expressed in roots [66]. The MADS-box TF AGL44 causes nitrogen-mediated morphological changes including rapid early seedling development, increased root capacity, and increased fresh weight of shoots [67]. In chlorophyll biosynthesis, protochlorophyllide oxidoreductases (PORs) catalyzes the reduction of protochlorophyllide to chlorophyllide via light stimulation; the TF RVE1 directly binds to PORA promoters, playing a crucial role in chlorophyll biosynthesis by triggering seedling greening during early plant development [27]. Collectively, HA up-regulates various TFs regulating seedling and root development, and abiotic stress tolerance either in the absence or presence of stress. 

Twenty TF genes were down-regulated by HA in DEG #2, and 44 TFs in DEG #3 (Figure 6d–f and Appendix A). Genes encoding Myb domain-containing TFs PAP1, MYB90 (AT1G66390), and MYB111 (AT5G49330), WRKY TFs WRKY30 (AT5G24110) and WRKY54 (AT2G40750), bHLH TFs MYC67 (AT3G61950) and AT4G20970, and ERF TFs TINY2 (AT5G11590) and CYTOKININ RESPONSE FACTOR 3 (CRF3, AT5G53290), C2H2 zinc finger TF AT5G60470, and MADS-box TF AT5G55690 were down-regulated by HA in both DEG sets (Figure 6d–f). Anthocyanin biosynthesis is controlled by a ternary TF complex consisting of WD40, bHLH, and Myb (WBM) TFs, such as TRANSPARENT TESTA GLABRA (TTG), GLABRA 3 (GL3), ENHANCER OF GLABRA 3 (EGL3), PAP1, PAP2/MYB90, MYB113, and MYB114 [68]. WRKY30 is involved in leaf senescence and responds to ROS, while WRKY54 participates in the regulatory network to regulate the leaf senescence process with possible cooperation with WRKY30 [69]. CRF5-overexpressing plants display smaller shoot size with reduced rosette leaf size and accelerated leaf senescence compared to wild-type plants but are tolerant to pathogen attack [70,71]. The physiological contributions of these commonly down-regulated TF genes overlap with GO terms of down-regulated DEGs, strongly suggesting that HA reduces the expression of genes involved in phenylpropanoid-related biosynthetic pathways triggering the reduction of anthocyanin and flavonoid accumulation and, furthermore, negatively regulating the expression of genes involved in plant growth.

## 3. Materials and Methods

### 3.1. Plant Materials, Growth Conditions and Treatments

To examine a salt tolerance assay in Arabidopsis, wild-type Arabidopsis (Col-0 ecotype background) seeds were surface-sterilized, prepared on 1/2 MS media (Duchefa Biochemie, Haarlem, the Netherlands) supplemented with or without 860 mg L^−1^ HA (Sigma Aldrich (Cat No#53680), St. Louis, MO, USA) [72], the most appropriate concentration of HA for conferring salt stress tolerance based on our previous report [11,14], in the absence and presence of 100 mM NaCl, and grown for 7 days. 

Italian ryegrass (*Lolium multiflorum* Lam. “Kowinearly”), seeds were kindly provided by the National Institute of Animal Science (Rural Development Administration, Korea). Twenty seeds were directly sown in potting soil No. 2 (Farmhannong, Korea), and germinated in the dark. Ten-day-old seedlings were treated with salt stress by submerging in 250 mM NaCl solution for 10 days, and water (as a control) or HA (860 mg L^−1^) sprayed onto the leaves at 0, 3, and 6 days after salt treatments. 

For transcriptome and qRT-PCR analyses, surface-sterilized seeds of wild-type Arabidopsis (Col-0) were grown on 1/2 MS media for 7 days, and seedlings were treated with salt (100 mM NaCl), 860 mg L^−1^ HA, or salt + HA for 9 h. All plants were grown at 22 °C under 16 h light/8 h dark cycle with 100 μmol photons m^−1^ s^−1^.

### 3.2. Chlorophyll Content

Freshly harvested plant samples extracted by 80% (*v*/*v*) acetone with agitation of 120 rpm in the dark for 1.5 days. The chlorophyll content was measured using a Beckman DU-800 UV/Vis spectrophotometer (Beckman Coulter, USA) at light wavelength of 645 and 663 nm and 80% acetone as a blank, and calculated as reported previously [73].

### 3.3. RNA Extraction, Illumina RNA-Seq and Analysis of RNA-Seq Data

Total RNA was extracted from triplicate biological replications using an RNeasy Plant Mini Kit (Qiagen, Hilden, Germany) according to the manufacturer’s instructions. Preparation of RNA libraries, mRNA purification, cDNA synthesis, cDNA library construction, Illumina sequencing, filtering, mapping, and DEG analysis were carried as described previously [12]. Mapping statistics for quality filtered reads generated for Arabidopsis samples are available in Appendix A and a previous report (for DEG #2). 

### 3.4. DEG and GO Analysis 

DEGs were identified based on a *q*-value threshold less than 0.05 for correcting errors caused by multiple testing [74]. GO enrichment analysis with the DEGs based on a log_2_ fold change ≥1 and ≤−1 was performed using agriGO v2.0 [75]. A GO-based trend test was performed through the Fisher’s exact test [76] to characterize the genes identified from DEG analysis. 

### 3.5. Validation by Quantitative Reverse-Transcription PCR (qRT-PCR)

Harvested samples as described above were frozen in liquid nitrogen immediately, and Total RNA extraction, cDNA synthesis, and qRT-PCR analysis were carried out as described previously [12]. qRT-PCR conditions were as follows: 95 °C for 5 min; 40 cycles of 95 °C for 30 s; 58 °C for 45 s; and 72 °C for 45 s; followed by 72 °C for 5 min. Melting curves were analyzed to confirm that the specific target was amplified. The relative expression levels were calculated using the comparative cycle threshold (ΔΔCt) method. The expression levels of target genes were normalized to the housekeeping gene, *TUBULIN* (*TUB*). Triplicate biological replications were performed. Primer sequences are available in Appendix A.

## 4. Conclusions

We conducted transcriptome analysis to understand the molecular mechanisms of HA promoting salt stress tolerance in Arabidopsis. Transcripts of genes related to responses to stimuli were significantly enriched, with up-regulation of diverse abiotic stress-related genes encoding HSPs and redox proteins by HA under salt stress conditions. By contrast, genes involved in biotic stress and secondary metabolic pathways, especially anthocyanin and flavonoid, were down-regulated by HA. HA also up-regulated various TFs regulating plant development and abiotic stress tolerance, and down-regulated TF genes involved in pigment metabolism and secondary metabolic processes. We concluded that HA triggers an overall alteration of gene expression involved in plant development and stress responses, and thus plays a role in helping plants tolerate salt stress. These findings provide transcriptome-scale molecular evidence for bioactivity driven by HA under salt stress conditions and will expand the environmental adaptability of HA to field crops.

## Figures and Tables

**Figure 1 molecules-26-00782-f001:**
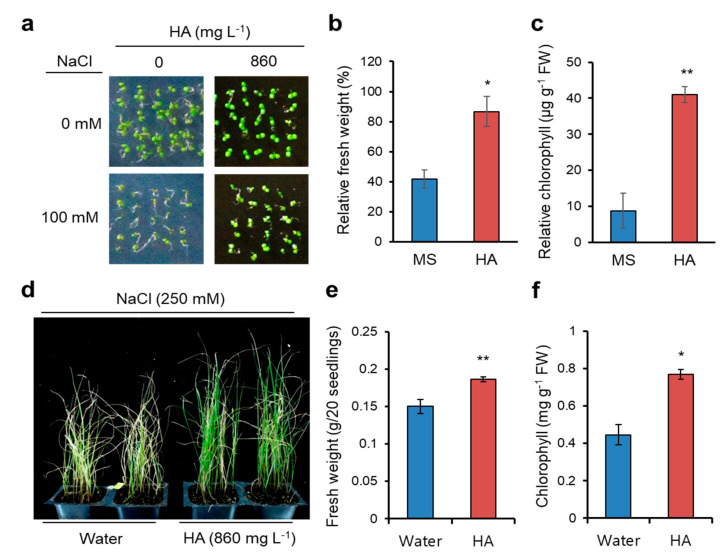
Humic acid (HA) confers salt stress tolerance in Arabidopsis and Italian ryegrass. (**a**–**c**) HA-induced salt stress tolerance in Arabidopsis. (**a**) Pictures shown 7-d-old wild-type Arabidopsis (Col-0) seedlings grown in 1/2 Murashige and Skoog (MS) media or that containing 860 mg L^−1^ HA supplemented with or without 100 mM NaCl for 7 d. (**b**) Relative fresh weight and (**c**) relative chlorophyll content (in the presence vs. absence of 100 mM NaCl). Fresh weight and chlorophyll contents were measured and relatively calculated (salt-treated values divided by non-treated). Data represent means ± SE, *n* = 3. * *p* < 0.05, ** *p* < 0.01 compared with seedlings grown in MS media. (**d**–**f**) HA-induced salt tolerance in Italian ryegrass. (**d**) Picture shown 20-d old Italian ryegrass seedlings exposed to 250 mM NaCl for 10 d with foliar application of water or HA (860 mg L^−1^) by spray. (**e**) Fresh weight and (**f**) chlorophyll contents were measured. Data represent means ± SE, *n* = 3. * *p* < 0.05, ** *p* < 0.01 compared with seedlings sprayed with water.

**Figure 2 molecules-26-00782-f002:**
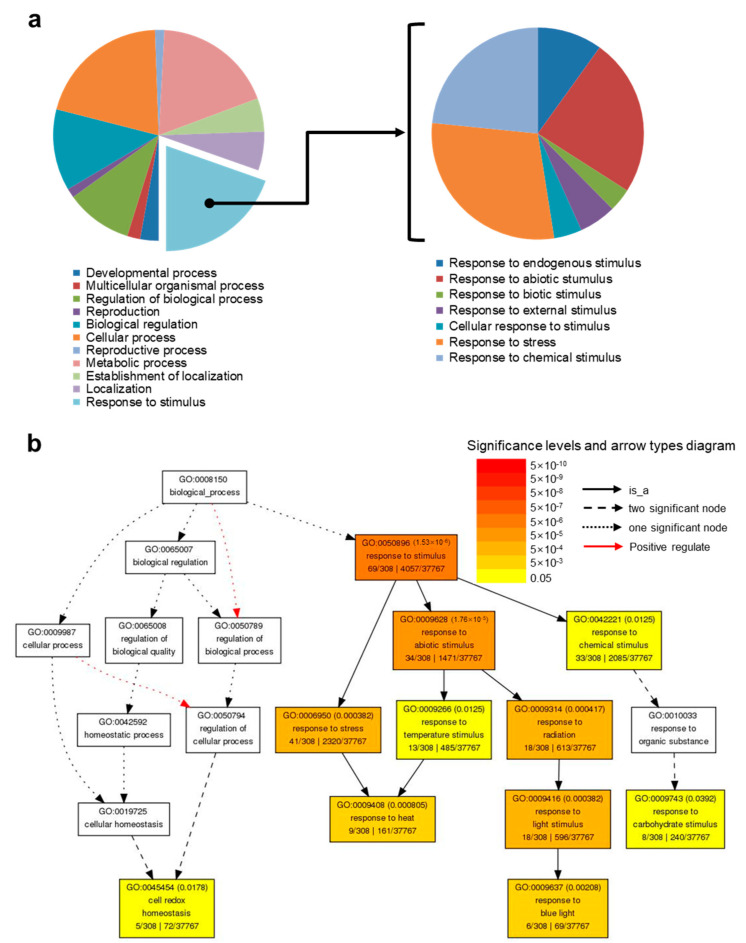
Gene Ontology (GO) enrichment analysis of up-regulated genes in differentially expressed genes (DEG) #3 (salt vs. salt + HA). (**a**) GO functional classification. “Biological processes” significantly enriched for up-regulated genes in DEG #3 (left) and sub-classification of DEGs involved in the response to stimulus (right). (**b**) GO term enrichment analysis for up-regulated genes in DEG #3. Each box indicates the GO term and description with the false discovery adjusted (FDR)-adjusted *p*-value; the color scale reflects these adjusted *p*-values. The fraction on the left side at the bottom is the number of genes in our dataset falling into that GO category out of the total number of genes in the list. Boxes with GO terms are presented hierarchically, with the root term at the top and child terms toward the bottom.

**Figure 3 molecules-26-00782-f003:**
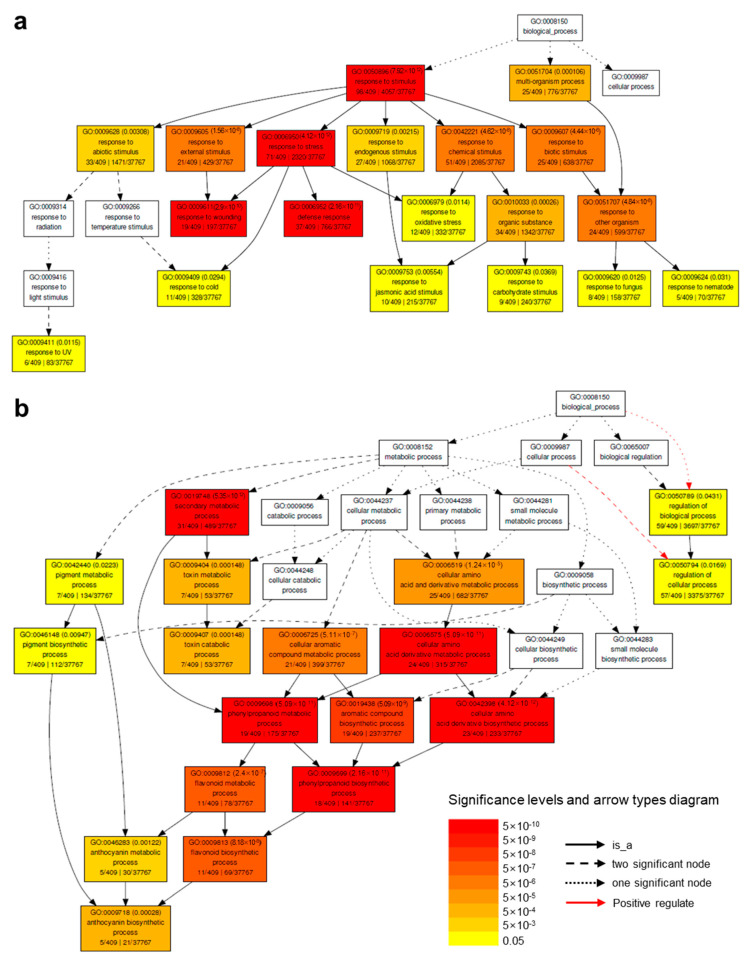
GO enrichment analysis of down-regulated genes in DEG #3 (salt vs. salt + HA). GO term enrichment analysis for down-regulated genes in DEG #3 enriched in the response to stimulus (**a**) and metabolic process (**b**) GO categories. Representations are described in the Figure 3b legend.

**Figure 4 molecules-26-00782-f004:**
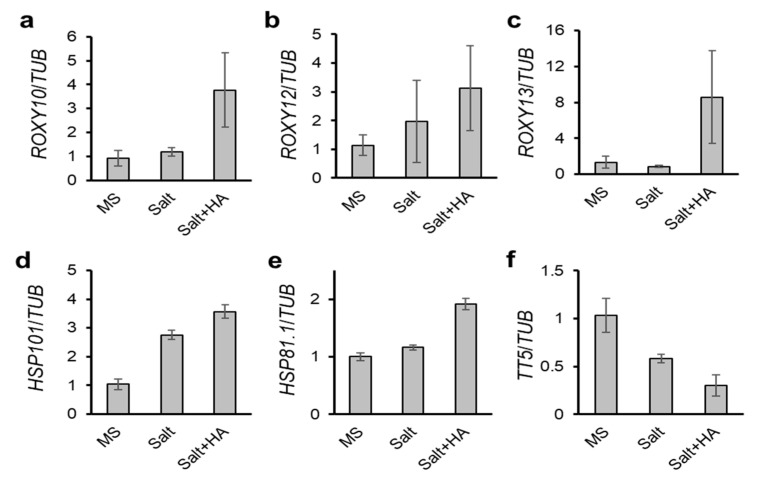
Validation of DEGs by qRT-PCR. One-week old Arabdopsis wild-type seedlings were treated with HA (860 mg L^−1^), salt (100 mM NaCl), or salt + HA for 9 h. Transcripts of ROXY10 (**a**), ROXY12 (**b**), ROXY13 (**c**), HSP101 (**d**), HSP81.1 (**e**), and TT5 (**f**) were validated by qRT-PCR and normalized to that of TUB. Data represent means ± SE, n = 3.

**Figure 5 molecules-26-00782-f005:**
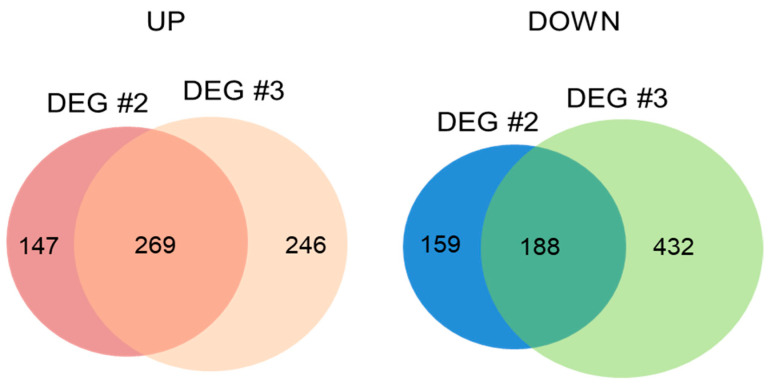
Venn diagram representing up- or down-regulated DEGs common to DEG #2 and DEG #3. The numbers represent significantly up- (left) and down- (right) regulated genes by HA in DEG #2 and DEG #3, indicating those regulated in common between DEG #2 and DEG #3.

**Figure 6 molecules-26-00782-f006:**
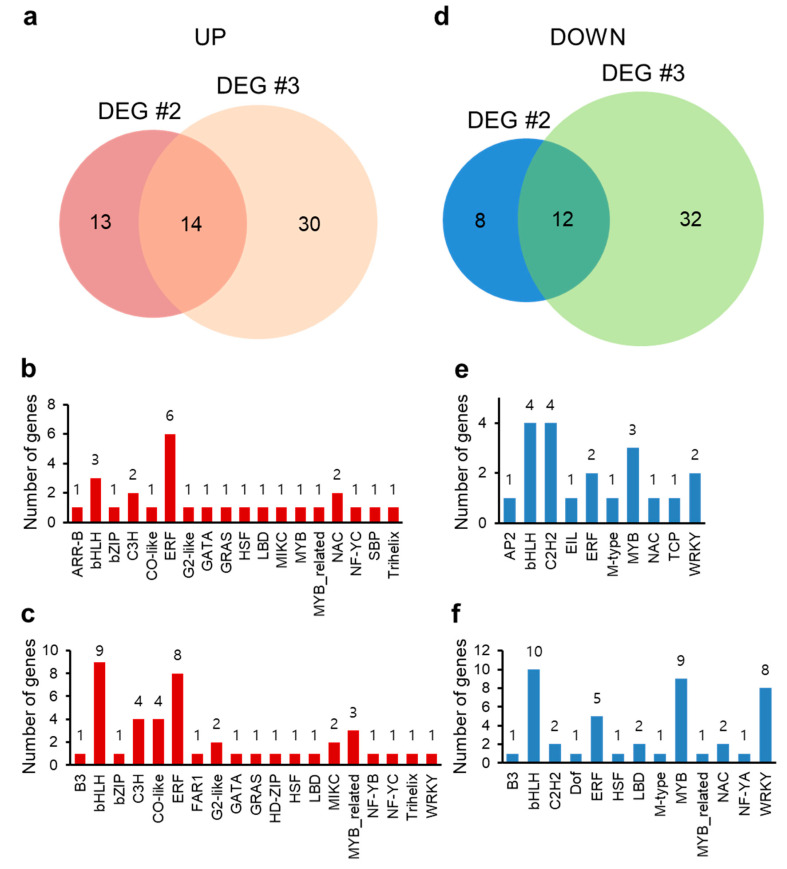
Number of up- or down-regulated transcription factor genes in DEG #2 and DEG #3. The number of TF genes significantly up- (**a**–**c**, log_2_ fold change ≥ 1) or down- (**d**–**f**, log_2_ fold change ≤ −1) regulated in DEG #2 (**d**,**e**) and DEG #3 (**c**,**f**) represented as a Venn diagram (**a**,**d**) or a bar graph (**b**,**c** for up-regulated TF genes; **e**,**f** for down-regulated TF genes).

## Data Availability

All data supporting the findings of this study are available in the main text or the Appendix A.

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
