# Peer review of "Transcriptome Changes Reveal the Molecular Mechanisms of Humic Acid-Induced Salt Stress Tolerance in Arabidopsis"

_molecules, 2021, doi:10.3390/molecules26040782_

Round 1
Reviewer 1 Report
The manuscript by Cha reported the transcriptome analysis of Arabidopsis treated by humic acid under a salt-stressed condition. Although the data presented in this report looks solid, unclear novelty and several mistakes seriously humpers this report's value.
1) The manuscript is submitted to the journal "Molecules" since the humic acid-induced tolerance of Arabidopsis was investigated. However, the Authors described no origin of humic acid used in this study anywhere in this report.
2) There is no novelty in the methodology of this research. "Gene ontology enrichment analysis of differentially expressed genes (DEG) extracted from transcriptome analysis of a small number of samples, with tiny follow-up confirmation experiments by RT-PCR" is a routine gene expression analysis.
3) Novelty is unclear. The transcriptome analysis result did not support any preexisted hypothesis and did not produce a novel hypothesis to be investigated in this study.
4) Some additional experiments are essential to claim some molecular mechanisms in HA induced tolerance.
Author Response
1) The manuscript is submitted to the journal "Molecules" since the humic acid-induced tolerance of Arabidopsis was investigated. However, the Authors described no origin of humic acid used in this study anywhere in this report.
Response) We described the origin of humic acid in line 441 of the original manuscript, but now we have moved to line 432 of the revised manuscript. A variety of HAs derived from different extraction sources are commercially available. In addition, detailed organic structures of HAs were proven to be versatile depending on the extraction sources. To avoid the use of HAs displaying different organic structures, we used one kind of commercial Sigma HA product showing the same lot number for all the experiments. Same as the response to Editor, comparative study of HA from different origin supports that Sigma HA derive from coal-related sources. We have added relevant references in Materials & Methods in the revised manuscript (line 432).
2) There is no novelty in the methodology of this research. "Gene ontology enrichment analysis of differentially expressed genes (DEG) extracted from transcriptome analysis of a small number of samples, with tiny follow-up confirmation experiments by RT-PCR" is a routine gene expression analysis.
Response) Transcriptome analysis provides powerful bioinformatics for comparative analysis in global transcription levels between treatments or genotypes. This study was also conducted to obtain evidence at the molecular level for various bioactivities of humic acid, especially on salt stress tolerance, which have not been reported to date. In addition, through transcriptome analysis, it was confirmed that the expression of genes encoding HSP and antioxidants that can induce tolerance to salt stress was increased in the humic acid-treated group. This suggests molecular-level evidence for the acquisition of salt stress tolerance of humic acid, which has not yet been reported, and also suggests the possibility that these genes could be molecular targets of humic acid.
3) Novelty is unclear. The transcriptome analysis result did not support any preexisted hypothesis and did not produce a novel hypothesis to be investigated in this study.
Response) As we answered the above question, no one other than us has attempted transcriptome approach to elucidate the molecular mechanisms of HA displaying various bioactivities. We also found that HA up-regulates abiotic stress-related genes, which help to endure against salt stress. Thus, we believe our approach and findings are novelty enough.
4) Some additional experiments are essential to claim some molecular mechanisms in HA induced tolerance.
Response) We were only allowed 10 days window for revision which is physically impossible to conduct additional experiments. We want reviewers to understand this.
Reviewer 2 Report
Why did you study the effects of HA on Italian ryegrass?
I highlighted a few grammatical mistakes and suggestions on the manuscript.

Author Response
Why did you study the effects of HA on Italian ryegrass?
Response) We are verifying the novel bioactivity of humic acid in rice, alfalfa, and Italian ryegrass, as food and feed crops, as well as in the model plant Arabidopsis. In particular, it was reported in the journal about the growth-promoting function of humic acid for alfalfa and Italian ryegrass. Therefore, the mechanism of salt tolerance has been verified in Italian ryegrass, one of the crop systems established in our laboratory.
I highlighted a few grammatical mistakes and suggestions on the manuscript.
Response) Thanks for your kind correction. We have modified all the issues in the revised manuscript.
Reviewer 3 Report
Dear colleges, I has been a pleasure to read your research paper and my recommendation is to be published as it is. However, I think you must correct (1) in the text, and the (2) is something I missed.
- Did you test different concentrations of HA? In the paper you used 860 mg L-1, for me sounds a quite specific concentration. If you tested different concentrations and this one was the only were the plants showed tolerance should be mentioned in the material and methods section.
- In this extensive analysis of genes and TFs, I missed a word explaining Salinity Overlay System genes AtSOS1-6 behavior and also a sodium uptake and movement inside the plants under HA and NaCl treatments.
Nothing else I really enjoyed this paper adding molecular data analysis for a well known biostimulant compound under abiotic stress conditions.
All the best
JCL
Author Response
Did you test different concentrations of HA? In the paper you used 860 mg L-1, for me sounds a quite specific concentration. If you tested different concentrations and this one was the only were the plants showed tolerance should be mentioned in the material and methods section.
Response) Yes, we performed a concentration screening experiment on various bioactivity of humic acid, and through a previous paper, it was found that 860 mg L-1 of humic acid treatment is the most appropriate concentration for the salt tolerance acquisition in Arabidopsis thaliana. In addition, it has been confirmed through prior studies that treatment with more than 2 mg L-1 of humic acid decreases the plant promoting function. Therefore, this concentration was used in the analysis of the present transcriptome and the experiment to obtain salt stress tolerance of Arabidopsis and Italian ryegrass. And according to the reviewer’s suggestion, the relevant content including references was added to Materials & Methods in the revised manuscript (line 432-433). Thanks for your suggestion.
In this extensive analysis of genes and TFs, I missed a word explaining Salinity Overlay System genes AtSOS1-6 behavior and also a sodium uptake and movement inside the plants under HA and NaCl treatments. Nothing else I really enjoyed this paper adding molecular data analysis for a well known biostimulant compound under abiotic stress conditions.
Response) I agree with your opinion. Transcriptome analysis is a good experimental tool to identify differences in a wide range of transcription levels between plants of different treatments or genotypes. Also, at the beginning of the experiment, changes in the expression of SOS genes, known as one of the key regulatory circuits in the mechanism of salt tolerance, were also predicted. However, in this transcriptome analysis, it was unable to observe changes in the expression of the genes encoding the SOS related proteins. Interestingly, in our previous experiments, we found that HKT1, a potassium transporter that plays an important role in the regulation of plant salt tolerance, is a molecular target of the HA-induced salt tolerance, and it is reported that it is regulated at the post-translational process level, not the transcription level (Reference no. 11). In addition, it was confirmed that SOS1 was not affected by HA.
We also reported on your other comment, salt absorption and transfer by HA treatment, in a previous paper. As a result of confirming the degree of absorption of salt into the plant in the HA and salt stress treatment groups through ICP analysis, the HA treatment did not prevent the absorption of salt ions into the plants (Reference no. 11). This fact suggests that the mechanism of salt tolerance of plants by HA treatment is not due to phytoremediation, but is regulated in various intracellular transcriptional and post-translational mechanisms. Therefore, in this study, we identified transcriptome change in the mechanism of salt tolerance by HA treatment, and genes encoding HSP and antioxidants (can inhibit protein aggregation and ROS accumulation, respectively) increased to reduce intracellular stresses caused by salt stress, rather than actually controlling the expression of salt transport genes.